# Challenges of Biomass Utilization for Bioenergy in a Climate Change Scenario

**DOI:** 10.3390/biology10121277

**Published:** 2021-12-06

**Authors:** Emanuelle Neiverth de Freitas, José Carlos Santos Salgado, Robson Carlos Alnoch, Alex Graça Contato, Eduardo Habermann, Michele Michelin, Carlos Alberto Martínez, Maria de Lourdes T. M. Polizeli

**Affiliations:** 1Department of Biochemistry and Immunology, Faculdade de Medicina de Ribeirão Preto (FMRP), University of São Paulo, Ribeirão Preto 14049-900, São Paulo, Brazil; emanuelleneiverthf@gmail.com (E.N.d.F.); alexgraca.contato@gmail.com (A.G.C.); 2Department of Chemistry, Faculdade de Filosofia, Ciências e Letras de Ribeirão Preto (FFCLRP), University of São Paulo, Ribeirão Preto 14040-901, São Paulo, Brazil; salgadojcs@hotmail.com; 3Department of Biology, Faculdade de Filosofia, Ciências e Letras de Ribeirão Preto (FFCLRP), University of São Paulo, Ribeirão Preto 14040-901, São Paulo, Brazil; robsonalnoch@usp.br (R.C.A.); eduardohabermann@gmail.com (E.H.); carlosamh@ffclrp.usp.br (C.A.M.); 4Centre of Biological Engineering (CEB), Gualtar Campus, University of Minho, 4710-057 Braga, Portugal; mimichelin.bio@gmail.com

**Keywords:** climate change, abiotic stress, cell wall remodeling, pretreatment, dedicated energy crop, biofuels

## Abstract

**Simple Summary:**

The most recent intergovernmental panel on climate change (IPCC 2021) has shown that the human influence on climate change has been unprecedented, predicting a global temperature increase of 1.5 °C in the earlies 2030s. The burning of fossil fuels has increased the emissions of nitrous oxide (N_2_O), methane (CH_4_), and carbon dioxide (CO_2_) to the atmosphere, amplifying the greenhouse effect in the last decades. In this scenario, the use of biorefineries, a renewable analog to petroleum refineries, has attracted a lot of attention since they use renewable sources as lignocellulosic feedstocks. However, climate change alters the temperature, rainfall patterns, drought, CO_2_ levels, and air moisture impacting biomass growth, productivity, chemical composition, and soil microbial community. Here, we discuss strategies to produce fuels and value-added products from biomass in a climate change scenario, potential feedstocks for bioenergy purposes, the chemical composition of lignocellulosic biomass, enzymes involved in biomass deconstruction, and other processes related to biomass production, processing, and conversion. Understanding these integrated factors involved in bioenergy production with plant responses to climate change shows that climate-smart agriculture is the only way to lower the negative impact of climate changes on crop adaptation and its use for bioenergy.

**Abstract:**

The climate changes expected for the next decades will expose plants to increasing occurrences of combined abiotic stresses, including drought, higher temperatures, and elevated CO_2_ atmospheric concentrations. These abiotic stresses have significant consequences on photosynthesis and other plants’ physiological processes and can lead to tolerance mechanisms that impact metabolism dynamics and limit plant productivity. Furthermore, due to the high carbohydrate content on the cell wall, plants represent a an essential source of lignocellulosic biomass for biofuels production. Thus, it is necessary to estimate their potential as feedstock for renewable energy production in future climate conditions since the synthesis of cell wall components seems to be affected by abiotic stresses. This review provides a brief overview of plant responses and the tolerance mechanisms applied in climate change scenarios that could impact its use as lignocellulosic biomass for bioenergy purposes. Important steps of biofuel production, which might influence the effects of climate change, besides biomass pretreatments and enzymatic biochemical conversions, are also discussed. We believe that this study may improve our understanding of the plant biological adaptations to combined abiotic stress and assist in the decision-making for selecting key agronomic crops that can be efficiently adapted to climate changes and applied in bioenergy production.

## 1. Introduction

There is a global concern about how plants and ecosystems will respond and adapt to new environmental conditions due to climate change. The effects of climate change on plants depend on the interaction of environmental factors essential for plant growth [1]. The increase in the concentration of carbon dioxide in the atmosphere would increase photosynthesis, which in turn contributes to increased plant growth [2]. However, the benefits to plant growth of increased carbon dioxide will probably be overcome by the deleterious impacts of drought [3] and heat stress [4]. Studies of interactions between species and environmental factors can contribute to understanding the adaptive responses of plants to future predicted conditions and the possible impacts of climate change on vegetation and biomass production.

Lignocellulosic biomass is a plant material that is receiving attention as a renewable resource to reduce dependency on fossil-based energy fuels and diminish biofuel feedstock costs [5]. The carbon dioxide captured during photosynthesis is used for biomass growth, and it is commonly balanced with the release of carbon dioxide from bioenergy/biofuel combustion [6]. Thus, the use of lignocellulosic materials does not increase the atmospheric carbon dioxide concentration compared to fossil fuels. However, the changes in climate conditions are projected to impact the growth, development, and yield of lignocellulosic biomass, and since they represent a promising alternative to generate clean energy and mitigate greenhouse gas emissions, their acclimation reactions to climate change should be explored [7].

The plant cell wall is constituted mainly by cellulose, hemicelluloses, and lignin. These components are intrinsically linked, forming a complex architecture that offers little access to the action of enzymes [8]. Therefore, cellulolytic, hemicellulolytic, and ligninolytic systems are necessary when thinking about the formation of fermentable sugars and the final production of bioenergy. In general, these systems are produced and secreted by microorganisms, mainly filamentous fungi, which have developed cellular secretory mechanisms. Several biomasses have been reported as potentially producing bioenergy, such as sugarcane bagasse [9,10,11,12,13,14,15], corn residues [16,17], paper sludge, and eucalyptus chips [18,19]. This review provides an overview of the impacts of plant responses to abiotic stress in cell wall properties and biomass digestibility, reporting the key challenges to be faced on the use of lignocellulosic feedstocks for bioenergy purposes in climate change scenarios.

## 2. Global Climate Changes: Evidence and Causes

The unequivocal influence of anthropogenic activities on Earth’s climate system has modified climate patterns worldwide. The intensification of greenhouse gases emissions such as carbon dioxide (CO_2_), methane (CH_4_), and nitrous oxide (N_2_O) in the last decades created an additional radiative forcing on climate, amplifying the natural greenhouse effect. As a result, long-term and significant warming trends have been observed since 1980, and the current temperature anomaly is set at approximately 1 °C above pre-industrial values. However, according to our most accurate climate models, the global average temperature is expected to keep increasing depending on future scenario emissions and may reach 2 °C if Paris Agreement goals are not achieved [20]. Furthermore, a warmer world will produce more intense and extreme weather events such as droughts, floods, and heatwaves, impacting natural and managed ecosystems in different world regions [21,22]. Therefore, there is urgency for studies that investigate how climate change will impact the biosphere.

The adverse consequences of climate change, as a result of the rise in anthropogenic greenhouse gas emissions, are among the main environmental concerns faced today on our planet. The increase in global average temperatures and changes in precipitation patterns are causing more intense extreme climatic events worldwide, which is convincing evidence that global climate change is already occurring, and its effects will become increasingly severe for all living beings [20].

Climate change is affecting natural and managed ecosystems in diverse ways. The multiple components of climate change are estimated to affect all levels of biodiversity [23], from organisms and biomes to societies and human economies [24]. As a result of climate change and extreme weather events, native and cultivated plants will face limited options to avoid habitat loss or extinction: adaptation, migration, or death [25]. These impacts will probably be more intense in developing countries where agriculture and livestock are the main economic activities. In 2009, a group of scientists led by Johan Rockström, from Stockholm University, reported in Nature [26] nine “planetary limits” or safe environmental limits that humanity must respect, and within which it can develop without the impacts caused to the environment being irreversible. Unfortunately, four planetary boundaries of humanity had already been transgressed: climate change, biodiversity loss, land-system change, and biogeochemical flows, while other planetary limits such as the use of freshwater, land-use changes, and ocean acidification were in danger of being overcome [27].

According to the most recent IPCC report (2021), human-induced climate change has led to an increased frequency and/or intensity of climate extremes in every region across the globe. The IPCC report also predicts that global warming will reach 1.5 °C in the early 2030s, and without reaching net-zero CO_2_ emissions—along with strong reductions in other greenhouse gases (GHG) such as CH_4_ and N_2_O, the planet will continue to warm. Various consequences of climate change will become irreparable over time, most particularly melting ice sheets, rising sea levels, biodiversity loss, and ocean acidification. Additionally, the impacts will continue to mount and compound as emissions of GHGs increase [20].

Land-use change due to agriculture is the principal cause of global deforestation, which causes a decrease in the volume of water transpired from plant leaves. In the Amazon Basin, the substantial deforestation observed in the last decades, combined with global climate change, is triggering more extreme drought. Besides that, as a result of deforestation and climate change, the carbon sink capacity of the Amazon Forest seems to be in decline. It would cause an abrupt shift from rainforest to savanna, with dangerous consequences for the entire planet [28]. The reduction in fossil-based energy consumption and the replacement with sustainable use of bioenergy from biomass are within the main options to fight climate change, reducing fossil carbon dioxide emissions to the atmosphere. However, biomass production and the potential use of biomass as a energy source are strongly dependent on environmental conditions and how plants respond to environmental changes.

## 3. Plant Responses to Climate Change

Biomass accumulation and its chemical composition are the final results of a complex set of metabolic pathways that respond to environmental changes [29]. It is widely accepted that increased atmospheric CO_2_ concentration, represented here as [CO_2_], improves plant growth and productivity by direct effects on photosynthesis and other physiological processes [30]. However, the C3 and C4 photosynthetic types of plants respond differently to higher [CO_2_]. Concerning C3 species, the higher [CO_2_] positively affects photosynthesis since in this condition, D-ribulose-1,5-bisphosphate carboxylase/oxygenase (Rubisco) speeds up carboxylation reaction and suppresses the oxygenation reaction. Consequently, the energy costs related to CO_2_ losses are reduced, leading to the accumulation of carbohydrates in leaves and improved biomass production [30]. Nevertheless, as photosynthesis in C4 species already occurs in saturated conditions of CO_2_ at the Rubisco active site, C4 crops would not benefit much from atmospheric CO_2_ increase [31]. However, some C4 species are also positively affected by increased atmospheric [CO_2_]. For example, Guinea grass (*Panicum maximum*) plants developed under 600 ppm of [CO_2_] showed an increased photosynthetic rate of approximately 25% when compared to plants growing under 400 ppm of [CO_2_] [32]. A similar response was reported for sugarcane (*Saccharum officinarum*) [33], in which photosynthesis improved under elevated [CO_2_], leading to higher biomass production. However, in general, the average increase in photosynthesis and productivity is still smaller in C4 plants compared to C3 species [30]. Moreover, this direct relationship between improved photosynthesis and enhanced biomass production is not straightforward since surplus carbon can be translocated to roots, flowers, fibers, and lignin synthesis [32]. In addition to the direct effects of high [CO_2_], photosynthesis and plant biomass production may be improved due to the effects on gene regulation [34], water relations [35], and enzymes [36].

Regarding high-temperature effects, the photosynthetic response of species depends on their optimum growth temperature and the intensity of warming. In general, biomass production and photosynthesis of C4 grasses under well-watered conditions are greatly improved under a warmer atmosphere [37]. This enhancement is related to higher enzymatic activities, chlorophyll synthesis, and changes in photosystems dynamics [37]. However, some species are susceptible to heat, and the global temperature increase might reduce photosynthetic efficiency due to stomatal closure resulting in minor CO_2_ flux into leaves through stomata and decreased Rubisco affinity for CO_2_ with rising temperature [38]. In addition, when the optimum growth temperature is exceeded, the production of reactive oxygen species is intensified, damaging biomolecules, and leading to the disruption of photosynthetic apparatus [39].

Human-caused climate change is modifying different chemical and physical aspects of the atmosphere, including temperature, air moisture, CO_2_ levels, drought, and rainfall patterns [20]. Therefore, plants will not respond independently to each one of these changing factors, but instead, plant biomass production will respond to a combination of atmospheric factors changing at the same time. In this scenario, unraveling the interactive effects of different climate change factors on biomass accumulation is critical for a deeper understanding of bioenergy production in the future. Unfortunately, multifactorial experimental designs are still scarce in the literature, and some uncertainty remains regarding plant responses to future climate conditions. Some examples reported for Guinea grass indicated that when plants develop under a warmer and CO_2_-enriched atmosphere, the positive effects of each factor are not additive, and production is not higher than under isolated effects of CO_2_ or warming [40].

Moreover, elevated [CO_2_] seems to offset part of the negative impacts of drought on plant performance since elevated [CO_2_] reduces stomatal conductance, transpiration, and water absorption from the soil, conserving soil moisture for more time [36,41]. Indeed, in maize (*Zea mays*), plants growing under elevated [CO_2_] exhibited improved photosynthesis only during periods of drought [42,43]. On the other hand, under a CO_2_-enriched atmosphere, switchgrass (*Panicum virgatum*) photosynthesis did not increase regardless of soil type [44]. Some experiments also indicate that increased temperature will negate any CO_2_ stimulation of photosynthesis and productivity in soybean plants (*Glycine max*) in the Midwest United States [45]. Studies that combine warming, drought, and elevated [CO_2_] together are even rarer, but some experiments suggest no interactions between all three factors in biomass production and photosynthesis of wheat plants [46,47]. Therefore, the use of plant biomass as a source of energy in future conditions of climate change needs to take into account all the future conditions of productivity and composition that will arise from these new environmental conditions.

## 4. The Role of Circular Economy in Mitigating Climate Change

Since the industrial revolution, the world has applied a linear economic model of “take, make, and dispose of” based on the presumption of abundant and inexpensive non-renewable resources [48]. However, total world energy consumption is predicted to increase 48% from 2012 to 2040, and the linear economic model needs to be replaced, attending to economic demands while preserving environmental needs [49]. The new bioeconomic model created in the last decades supports the reuse and recovery of natural resources as an alternative to fossil and non-renewable resources [48]. For this purpose, global research programs have been encouraged to discover new and sustainable energy supply as the global economy can no longer depend on fossil fuels that release considerable amounts of greenhouse gases [50].

The realization of the circular bioeconomic model is based on the biomass-based biorefinery, in which abundantly available and renewable lignocellulosic biomass is converted into fuels and chemicals [48]. The biorefinery concept developed a path forward to a society less dependent on fossil fuels. In addition, it contributed to mitigating climate change, mainly because it is considered to maintain net-zero CO_2_ emission into the atmosphere because the CO_2_ generated through the use of the resources is used in biomass production via photosynthesis [6].

A biorefinery represents the renewable equivalent of a petroleum refinery, but with the possibility to use the renewable lignocellulosic feedstocks to produce novel value-added chemicals and fuels that are otherwise not obtainable from fossils (Figure 1) [51]. The International Energy Agency Bioenergy Task 42 defines biorefining as “the sustainable processing of biomass into a spectrum of marketable bio-based products (chemicals, materials) and bioenergy (biofuels, power, heat)” [52]. Nonetheless, achieving this goal requires the selection of microorganisms capable of utilizing biomass as substrate, a deep understanding of the biomass chemistry, and of the environmental effects on the bioenergetic potential of feedstocks, which are crucial for establishing biorefinery systems [6].

Considering the energy content (USD/GJ), lignocellulosic materials have their cost estimated to be 50% lower than another feedstock, as crude oil, natural gas, corn kernels, and soy oil [50,53]. The high production and low cost of lignocellulosic biomass confirm its potential as an abundant source for bioenergy generation. In terms of economic evaluation, fine chemicals and other commodities that are derived from lignocellulosic biomass are the ones with the highest potential to maximize the value of the bioenergetic chain since the products can be applied in various economic sectors (e.g., pharmaceutical, petrochemical, construction, automotive, cosmetics, agroindustry, and others). At the second valuation are the biofuels and materials, followed by energy commodities, fertilizers, and pesticides [54].

Second-generation or cellulosic ethanol is the biofuel with the highest potential to substitute fossil fuels since it is possible to increase the biomass yield without altering the area used to cultivate the feedstocks by using previously discarded plant wastes. Considering the worldwide basis, the cellulosic ethanol market is predicted to achieve 27 billion gallons/year by 2022, translating the world’s strong demand for second-generation ethanol [55]. In addition, a couple of other products produced in biorefinery have significant interest due to their commercial uses, such as succinic, fumaric, malic, and glutamic acid, glycerol, sorbitol, and xylitol/ arabinitol, which nowadays are formed through the high-value replacement products [51].

## 5. Potential Feedstocks for Bioenergy Purposes

The production of the first generation (1G) ethanol and biodiesel raised a concern about fuel vs. food conflict since it depends on the utilization of food crops to produce sucrose, starch, grains, and vegetable oils, impacting food supply and land sustainability [56]. Differently, billions of tons of lignocellulosic materials are produced every year around the world; however, most of these residues are burned or discarded, while it could be converted to second-generation (2G) ethanol and other value-added products due to the rich carbohydrate fraction [6,48]. The main advantage of 2G ethanol over first-generation fuels is that the former does not compete for fertile lands used in food crop production [57].

Although lignocellulosic materials are potential sources for the production of 2G fuels, some factors threaten the economic viability of biomass preprocessing and must be considered when choosing a feedstock for bioenergy production, such as (I) biomass availability; (II) the nutrient input during the growing season; (III) the logistics of biomass collection, storage, and transportation; (IV) chemical composition; (V) calorific value; and (VI) potential ethanol yields [56,57]. Moreover, biomass employment as an energy source will depend on considering these factors to a particular region since most of them are influenced by the climate and growing conditions [58].

Furthermore, lignocellulosic feedstocks used in biorefineries can be classified into primary categories: (1) residues and wastes and (2) dedicated energy crops. Residuals and agricultural wastes are mainly derived from household practices, manufacturing, and agriculture [59]. Residue materials include organic matter, plastics, and municipal solid wastes, which are most useable due to the methane stored in the material. Furthermore, they have a heterogeneous nature, frequently containing plastics, metals, and glass, along with high transportation costs, making difficult its use as a biorefinery feedstock [59]. Agriculture wastes include residues generated after crop harvest or in the processing of food crops [54]. The most important cellulosic feedstocks are sugarcane bagasse, corn stover, wheat, rice, barley straw, and sorghum stalks [8,11,16,17,54]. All of them are produced in considerable amounts and have the advantage of being made along with the food crop without requiring additional costs. However, part of the biomass must be left in the field for soil incorporation, reducing the amount available for bioenergy purposes [54,59].

Dedicated energy crops are cultivated for energy conversion processes without replacing food crop production. They can be classified into herbaceous and short-rotation woody crops [58,59]. Short-rotation woody crops include soft and hardwoods with a harvest cycle of five to eight years, being eucalyptus and poplar the main species with bioenergy potential. Herbaceous energy contains little woody content and represents an important alternative for diversification of feedstock, helping to expand the energetic matrix beyond the agriculture wastes [58,59]. Perennial grasses, such as *Miscanthus* and switchgrass, are widely applied as energy crops in Europe and the United States, but other grassy biomasses also have demonstrated significant potential as bioenergy crops like the ones from the *Panicum* genus, which has high biomass production (30 ton/ha) and great ethanol yield 8571.0 L/ha [50,60].

The use of perennial grasses for biofuel and biorefinery purposes has some advantages as high biomass yield, broad geographic adaption, low lignin content, easy hemicelluloses degradation, and production during the intercropping period to avoid intermittent biofuel production [50,61]. Moreover, they have low mineral-nutrient inputs making it possible to grow in lands abandoned for agriculture uses, not requiring extensive capital-intensive processes, making the final product economically feasible [62].

## 6. Chemical Composition of Lignocellulosic Biomass

Lignocellulose is a material synthesized by plants to form a thick layer named the cell wall, which is present in all cell types protecting against pathogens [50]. Polymeric carbohydrates (cellulose and hemicellulose) and lignin are the primary constituents of lignocellulosic biomass with minor amounts of proteins, pectin, extractives, and inorganic compounds. Nonetheless, the ratios among cell wall components and their structure will depend on several factors such as age, species, and culture conditions [63].

Cellulose represents 25–50% of total lignocellulosic dry matter, while hemicellulose represents 20–40%, and lignin constitutes 15–25% of the entire content [64]. The cellulosic component is composed of linear β-1,4-glucan chains that are tightly packaged into microfibrils and generally divided into two regions, one with low molecular order (called amorphous cellulose) and the other with high crystalline order (called crystalline cellulose). Likewise, the polymerization degree in cellulose is the highest among the lignocellulosic polymers and could range from 100 to 10,000 depending on its source, being responsible for cellulose’s low flexibility [65].

Hemicellulose polymers are branched heteropolysaccharides mainly composed of residues of xylose, arabinose, glucose, and mannose, which hold some functional groups such as methyl, acetyl, glucuronic, and galacturonic cinnamic acids [66]. Van der Waals interactions and hydrogen bonds are the main forces between cellulose microfibrils, whereas hemicelluloses bind cellulose fibrils to the surface through non-covalent linkages [63]. Lignin is produced to provide structural reinforcement to the plant tissue, and it consists of a phenylpropanoid polymer mainly derived from three monolignols units: p-coumaryl alcohol (H), coniferyl alcohol (G), and sinapyl alcohol (S), which may differ between species and cell tissue type. The differences in monolignols composition substantially affect the deconstruction of biomass and delignification [66]. Meanwhile, lignin polymeric structures are assumed to arise from the polymerization reaction of phenoxy radicals formed by oxidative enzymes in the cell wall [67].

The cell wall constituents are intertwined with each other forming a complex and recalcitrant structure. Zhang [68] found that only 0.0023–0.041% of the β-1,4-glycosidic bond of cellulose is accessible for enzymatic degradation. Hemicellulose is easier than cellulose to hydrolyze, whereas lignin polymer is resistant to hydrolysis and cannot be fermented to produce ethanol, being used in chemicals and resins production as part of the biorefinery process [50,64].

Furthermore, ferulic acid, generated through the phenylpropanoid pathway, can couple oxidatively with lignin and, due to its carboxylic group, can esterify arabinose residues from hemicelluloses establishing lignin carbohydrate complex (LCC) (Figure 2) [69]. Thus, the formation of LCCs has an excellent contribution to cell wall recalcitrance against biological and chemical degradation. Other possible reasons for this resistance are (I) low accessibility of crystalline cellulose fiber that prevents cellulase attack and is also reported to have a role in cellulase adsorption and processivity and (II) the presence of hemicellulose and lignin on the surface of cellulose blocks the enzymatic access to the substrate, which is required for bioconversion of cellulose to glucose [70,71].

## 7. Pretreatment

The low saccharification rate of cellulose in lignocellulosic biomass is mainly promoted by the steric hindrance imposed by lignin, hemicellulose, and crystalline morphology of cellulose. Usually, in the native form, less than 2% of the polysaccharides from lignocellulosic biomass are hydrolyzed by enzymes. Therefore, pretreatment is a crucial step in the lignocellulosic-based industry to reach a high conversion of cell wall polysaccharides into fermentable sugars [50].

Nonetheless, pretreatment is the most energy-demanding step in biomass conversion and can contribute to more than 40% of the total processing [72]. This challenge has encouraged the investigation for technologies to achieve scalable, efficient, and greener pretreatments. The main goals for biomass pretreatment are to remove and preserve the hemicelluloses and lignin and to reduce the crystallinity index of cellulose and increase the cellulases accessibility [73].

Generally, the pretreatment methods are divided into four categories: physical, chemical, physicochemical, and biological (Figure 3). Combinations of two or more methods have also been described in the literature. Nevertheless, it is difficult to define an ideal pretreatment, although some examples are reported in literature according to hemicelluloses or lignin removal and cellulose conversion (Table 1). An effective pretreatment should allow high carbohydrates recovery; avoid or limit the formation of inhibitors or sugar degradation; enable to value all compounds present in lignocellulosic materials, not only cellulose; minimize energy input; employ green solvents; limited generation of wastes; and be cost-effective [74,75].

Among the factors that could influence pretreatments effects on biomass that are usually considered in the severity factor of the pretreatment (temperature, time, and pH), the pH significantly affects this step [76]. At low pH, most hemicelluloses are removed from the solid material and released as monomeric sugars, while cellulose is kept almost intact. Working at neutral or around neutral pH leads to partial hemicelluloses hydrolysis due to organic acid formation resulting in the autohydrolysis process; still, most hemicellulose stays in oligomeric form since the conditions are not severe enough. In contrast, lignin is dissolved at mild acidic (pH 5) and alkaline pH, while most cellulose and hemicellulose are retained in the solid fraction [51]. Consequently, the selection of the pretreatment method depends on the final application.

The employment of ligninolytic enzymes in biological pretreatment is a potential alternative to the pretreatment step since it does not require high energy demand and no chemical is applied in the process; however, their production costs need to be considerably reduced [77]. The on-site production of ligninolytic enzymes is an alternative to reduce production and operation costs significantly [78]. Metabolic engineering of microbial strains of industrial interest constitutes a tool that modifies the protein expression and regulation, increasing the yield of ligninolytic enzymes. Some examples of successfully engineered strains to improve ligninolytic enzyme production are discussed in Li et al. (2016a, 2016b) [79,80].

**Table 1 biology-10-01277-t001:** Effect of different pretreatments on the breakdown of lignocellulose biomass and the enzymatic conversion of cellulose into glucose, with emphasis on grasses.

Biomass	Pretreatment	Details	Conditions	Maximal Removal (%)	Glucose Yield (%)	Reference
*Miscanthus* (Mx2779 and Mxg)	Steam explosion		180 (9 bar), 200 (15 bar), 210 (20 bar), and 225 °C (25 bar) for 5, 10, or 15 min	73.2% xylan (Mx2779) 78.9% xylan (Mxg) 200 °C, 15 bar, 10 min	68% (Mx2779) 41% (Mxg)	[81]
(1) Switchgrass (2) Corn stover (3) *Miscanthus*	Microwave-assisted DES	ChCl:lactic acid (1:2)	45 s, 800 W (152 °C)	(1) 83.7% xylan 72.2% lignin (2) 90.1% xylan 79.6% lignin (3) 77.5% xylan 65.2% lignin	(1) 75% (2) <40% (3) 78.5%	[82]
Switchgrass	DES	ChCl:glycerol (1:2) with 20 wt% water additions	120 °C for 1 h	85.35% xylan 56.82 lignin	89%	[83]
Elephant grass (leaf, stem, and whole plant)	Acid	H_2_SO_4_	5, 10, or 20% H_2_SO_4_ at 121 °C for 30 min	85.02% hemicellulose from leaf (20% H_2_SO_4_)	89.2% (leaf, 20% H_2_SO_4_)	[84]
(1) Elephant grass (2) Sugarcane bagasse	High-pressure CO_2_/H_2_O		180, 200, or 220 °C with a constant initial CO_2_ pressure of 50 bar	(1) 59.2% xylan (2) 46.4% xylan	(1) 77.2% (2) 72.4 (220 °C)	[85]
*Miscanthus*	Biological	Bacteria (laccase)	37 °C, 200 rpm, 96 h (with a mediator)	59.5% lignin (*Pseudomonas* sp.)	87%	[86]

DES: deep eutectic solvents; ChCl: choline chloride; SF: severity factor.

## 8. Enzymatic Deconstruction of Lignocellulosic Biomass

After the pretreatment step, the biomass needs to be broken down into monomeric (C5 and C6) sugars that can be converted into ethanol or other chemicals through the fermentation process. The degradation of lignocellulosic material into simple sugars can occur either enzymatically or chemically. Although the enzymatic process is still under development to achieve economic feasibility, it has been shown to be the best choice over the past few years as it is, by nature, a more specific and ecological process. In this way, by presenting milder operating conditions, there is a reduction in the formation of inhibitory biological compounds [10,15,87].

The high complexity and association of the carbohydrate–lignin complex in the plant cell walls is the main obstacle in the bioconversion of lignocellulosic materials into fermentable sugars, requiring diverse enzymes with different functions Although microorganisms, especially fungi, are great enzyme producers, the enzymatic extracts from a single microorganism do not have all enzymes necessary to degrade cellulosic materials optimally, considering the conversion rate greater than 70% in 48 h of hydrolysis with more than 10% solids load in low enzyme load [73].

Consequently, adding extracts from various microbial sources is essential to improve enzymatic hydrolysis efficiency [88]. The fungal strains with the greatest industrial interest for cellulolytic enzymes are the genera *Trichoderma*, *Penicillium*, and *Aspergillus*. They are described as producing the main cellulolytic enzymes such as cellobiohydrolases, endoglucanases, and β-glucosidases, and other essential enzymes for biomass deconstruction [89].

For the conversion of lignocellulosic biomass into fermentable sugars to occur in a viable way, some challenges must be overcome for commercial applications, such as (I) reduce costs of enzyme production; (II) selection of suitable enzyme sets with the optimal amount of each enzyme to optimize synergistic effects; (III) enzymes that can function effectively at high solid biomass loads; (IV) improvement of enzyme catalytic efficiency; and (V) reduction of the inhibition of the β-glucosidases final product [78,90].

Furthermore, the composition and structure of biomass have a considerable influence on enzymatic hydrolysis and can be altered under future climate conditions. Therefore, knowing the effects of climate on biomass structure, along with choosing the pretreatment category to be applied, are essential parameters to be considered for the development of the enzymatic cocktail [73].

The enzymes that should be composing enzymatic cocktails necessary for the complete hydrolysis of lignocellulose feedstock are core cellulolytic, hemicellulolytic, and accessory enzymes. These enzymes are included in different CAZy families and are discussed below (Figure 4).

### 8.1. Core Cellulolytic Enzymes

At least three classes of cellulolytic enzymes are essential for the enzymatic hydrolysis of cellulose: endo-β-1,4-glucanases (EC 3.2.1.4), cellobiohydrolases, or exoglucanases (EC 3.2.1.91) and β-glucosidases (EC 3.2.1.21)). The cellobiohydrolases are composed of two families: cellobiohydrolases I and II, which act on the β-1,4-glycoside bonds at the reducing and non-reducing ends of the cellulose, respectively, releasing cellobiose. Endoglucanase (EC 3.2.1.4) hydrolyzes β-1,4-glycosidic bonds in amorphous regions of cellulose, while β-glucosidases (EC 3.2.1.21) act on cellobiose and cellodextrins to release glucose. β-glucosidases play a key role as catalysts in lignocellulosic degradation since this enzyme releases glucose, used by *Saccharomyces cerevisiae* in the fermentation process for the production of ethanol, for example [91].

### 8.2. Hemicellulolytic Enzymes

Since hemicellulose is a hetero and branched polymer composed of different monomeric units and functional groups, its debranch and degradation to C6 and C5 sugar need the cooperation action of numerous enzymes (Figure 4). These enzymes assist in hemicellulose removal and increase the effectiveness of cellulase’s attack by exposing the cellulose microfibrils [92].

The enzymes endo-β-1,4-xylanase and (EC 3.2.1.8) and β-D-xylosidase (EC 3.2.1.37) are the most important xylan-degrading enzymes, especially for arabinoxylan (AX) from grasses. While endo-β-1,4-xylanase cleaves glycosidic linkages in the internal part of the xylan backbone, releasing xylose and xylooligosaccharides, the β-D-xylosidase enzyme hydrolyzes β-1,4-D-xylans, xylooligosaccharides, and xylobiose from nonreducing ends, releasing xylose as the product [73,93].

The hydrolyze of L-arabinoses units from heteropolysaccharides requires the action of arabinases (EC 3.2.1.99) and arabinofuranosidases (EC 3.2.1.55). Arabinases act in the α-1,5-arabinofuranosidic bonds between arabinose residues present on arabinan to release arabinose in mono or oligomeric form [94]. Arabinofuranosidases hydrolyze, from the nonreducing end, the α-1,2, α-1,3, and α-1,5- glycosidic linkages in arabinan, arabinoxylan, and arabinogalactan to arabinofuranosidic residues [95].

As previously discussed, glucuronic acid residuals are also part of hemicelluloses, as glucuronans and glucuronoglycan. β-glucuronidases (EC 3.2.1.31) hydrolyze glucuronic acid [96].

Xyloglucan endo-β-1,4-glucanases (EC 3.2.1.151) involve the hydrolyze of xyloglucan hemicellulases, consisting of a β-1,4-glucan backbone with xylosyl side chains linked at O-6 position of glycosyl residue [91].

Finally, for the deconstruction of mannan hemicelluloses, which are formed by a linear backbone of D-mannose/glucose with galactosyl side groups, endo-1,4-β-mannosidase (EC 3.2.1.78) and β-mannosidases (EC 3.2.1.25) are needed. The former hydrolyzes linkages in the mannans’ backbone, whereas the latter hydrolyze mannans from the nonreducing end [97].

### 8.3. Accessory Enzymes

Besides the enzymes mentioned above, the enzymatic cocktail should also cover accessory enzymes such as feruloyl and acetyl esterases and lytic polysaccharides monooxygenases, and proteins, such as expansins, to completely degrade lignocellulose substrates. These enzymes have been described to assist biomass deconstruction by opening up the lignocellulosic matrix and acting synergistically with the canonical hydrolytic enzymes, allowing extraction of more bioenergy power from lignocellulosic biomass and helping to reduce the costs [98].

Among auxiliary activity enzymes, AA9 (formerly GH61), a lytic polysaccharide monooxygenase (LPMO), has been reported mainly in the fungal system. The AA9 is copper-dependent and cleaves cellulose chains by an oxidative mechanism either in C1 or C4 carbon of glucose in the cellulose chain, requiring an electron donor and molecular oxygen [99]. Fungi also secrete a diversity of carbohydrate-specific oxidoreductases classified in auxiliary activity family 7 (AA7). The most known are cellobiose dehydrogenase and cellooligosaccharide dehydrogenase (CDHs), catalyzing the oxidation of the reducing end C1-OH in cellooligosaccharides and cellobiose to the corresponding lactones, donating electrons to LPMOs in the process. Lactose, xylooligosaccharides, and chitooligosaccharides could also be converted to lactones by the enzymes of the AA7 family [98]. Furthermore, the combined action of AA7, AA9 with hydrolytic enzymes has been demonstrated to result in higher levels of sugar release since AA9 acts creating reactive sites for cellulases on the recalcitrant crystalline regions, with low levels of monomeric sugars released [100].

Acetyl xylan esterase (EC 3.1.1.72) catalyzes the deacetylation of xylans and xylooligosaccharides by removing *O*-acetyl substitutes from the C-2 and C-3 positions [101]. Feruloyl esterases (EC 3.1.1.73) act on carboxylic ester linkages between arabinose and ferulic acid side groups forming the lignin–carbohydrate complexes (LCCs). As previously described, LCCs significantly reduce biomass recalcitrance by hiding cellulose microfibrils to the hydrolytic attack of CAZy enzymes [97].

Another class of accessory enzymes is the non-hydrolytic/non-oxidative proteins, known for their amorphogenesis-inducing action. These enzymes disturb the plant cell wall structural matrix, facilitating the deconstruction of the polysaccharides by hydrolytic and oxidative enzymes [102]. Swollenin is an example of such a protein that targets the amorphous regions of cellulose, weakening hydrogen bonding with hemicellulose rather than directly disrupting the crystalline regions [103]. Therefore, by promoting substrate amorphogenesis, swollenin increases plant cell wall porosity and provides the catalytic enzymes enhanced access to the glycosidic linkages within the sugar polymers.

Laccases, lignin peroxidases, and manganese peroxidases are also accessory enzymes responsible for lignin depolymerization and chemical modification [104]. Generally, it is preferred to be added before enzymatic hydrolysis during biological pretreatment of biomass to avoid the formation of lignin-derived compounds that could impact the enzymatic hydrolysis of cellulose and fermentation [105].

## 9. Challenges of Biomass Utilization for Bioenergy in a Climate Change Scenario

The agricultural industry and climate change are uniquely dependent on each other since the abiotic and biotic stresses caused by environmental changes have adverse effects on plants. The increment of average temperature, heat waves, change of CO_2_ atmospheric levels, variations in annual rainfall, and drought, distress plant development, and yield, comprising biochemical, physiological, molecular, and morphological modifications of plants [1,106].

In the climate change scenario, lignocellulosic biorefinery could be strongly impacted since climate change has been described to affect biomass growth, productivity, chemical composition, and soil microbial community [1,107,108,109,110] (Figure 5). Therefore, since the major feedstocks for lignocellulosic biorefinery are dedicated energy crops (e.g., switchgrass, *Miscanthus giganteus*, Guinea grass, energy cane, and others), or plant residues (e.g., sugarcane bagasse, corn (maize) stover, wheat straw, and others), climate change and the abiotic stress caused by them might significantly impact the production of cellulosic ethanol and other value-added commodities by reducing the yield and availability of these sources of biomass, along with changes in metabolic pathways.

Among climate changes, drought and warming have the greatest negative effect on crop yield [111]. Results from different methods consistently exhibited a negative effect of the rise in global temperature on maize and wheat yield. Without the effect of CO_2_ fertilization, genetic improvement, and effective adaptation, each degree-Celsius increase in global mean temperature would reduce global yields of maize by 7% and wheat by 6.0%, on average [7,107,112]. In addition, as sugarcane is a relatively high water-demanding crop, water deficiency can lead to up to 60% in productivity losses [109].

Furthermore, recent evidence has shown that combined stresses might affect plant metabolism differently from isolated stress, showing unique responses [1,110]. As climate changes will be experienced as combined stresses, their effects under real field conditions must be studied. Combined effects of elevated temperature and CO_2_ enrichment showed that photosynthetic rates of maize are not responsive to increased levels of CO_2_, due to the concentrating mechanisms of C4 plants. However, as CO_2_ enrichment leads to improved water relations, the growth rates of maize can be positively affected under CO_2_ elevated concentrations, but this positive effect is much smaller in C4 than for C3 plants [113]. Furthermore, if temperature increase in levels higher than optimum plant growth temperature, CO_2_ enrichment becomes deleterious and can exacerbate the negative effects of elevated temperature since the stomatal closure induced by CO_2_ contributes to elevating leaf temperature due to reduced evapotranspiration rates [114]. In this case, elevated temperature and CO_2_ enrichment would result in lower maize yields [110,114].

Hatfield et al. (2011) [113] concluded that future increases in temperature and CO_2_ would result in a minimum of 3% decrease in maize under well-watered conditions. Even for C3 plants, such as wheat and eucalyptus, the beneficial effects of CO_2_ enrichment on carbon fixation and yields diminish at elevated growth temperatures and might disappear at intensely elevated temperatures [114].

Considering the combined stress of drought and elevated temperature, the effects of drought outweigh the effects of temperature and the stomatal remain closed, leading to reduction of wheat and maize photosynthesis rate, plant length, leaf area, total dry weight, and yield [115,116]. Moreover, under warming and water stress the C4 Guinea grass maintained the leaf biomass production in similar levels of current environment conditions (control group) [117].

Likewise, when Guinea grass was developed under warming and CO_2_-enriched atmosphere (eTeC), although elevated CO_2_ reduced the foliage by favoring biomass partitioning to steam and accelerated leaf maturation, the effect of elevated temperature partially offset CO_2_ negative effect and led to comparatively high leaf production in eTeC [118]. Leaf dry Guinea grass biomass was found to be 42% higher under eTeC conditions, possibly due to the stimulus of starch exportation and the carbon surplus provided by enhanced photosynthesis [32]. In addition, although Guinea grass productivity is expected to rise under eTeC conditions in well-watered conditions [119], the nutritional requirement for Ca, N, and S are also expected to increase, which can lead to an increase in fertilizers and harvesting costs [40].

Furthermore, plants belong to an ecosystem interacting with multiple microorganism communities, which are also affected by climate change. Studies have shown that climate changes would affect soil microbial diversity, abundances, and activities [120,121,122]. Oliveira et al. (2020) [123] studied the impact of warming and drought on fungal communities showing that under these conditions some phytopathogenic fungi, like *Curvularia*, *Albifimbria,* and *Fusarium* species were more abundant. Some of these species are correlated with seedling’s death, grain and seedling rot, and reduced growth in a variety of host crops [124]. Furthermore, *Fusarium* and *Albifimbria* genus are widely recognized as nitrous oxide (N_2_O) producers, a greenhouse gas described as much more dangerous than CO_2_ for the environment [125,126].

The way that plants are going to react to climate changes is difficult to predict. The crop species, the intensity, and durability of stresses, the geographic location, the optimum temperature of growth, the plant cultivar, are factors that influence plant response [1,112]. Therefore, experimental studies with a combination of stresses that simulate a realistic future climate scenario for major crops will be crucial and strategic to predict the gains or losses that humanity will have from global climate change.

The aforementioned crops, such as maize, sugarcane, and wheat are widely cultivated for food supply, representing important sources that can be used as sources of lignocellulosic biomass, like sugarcane bagasse, wheat straw, and corn stover. Therefore, the reduced yield derived from plant response to climate change might impact both food and bioenergy supply. In this sense, the use of dedicated energy crops as forage grasses represents an important alternative to bioenergy production since these plants seem to have high optimum growth temperature and several mechanisms of adjustment under abiotic stress induced by expected futuristic climate conditions, resulting in higher yields in the future climate scenarios and other advantages mentioned in Section 5. However, increased biomass accumulation of grasses under warming is strongly dependent on soil moisture levels and can be nullified with drought spells [117].

Despite the yield, several factors must be considered regarding climate change since environmental conditions might affect the composition of the plant’s organic matter, impacting the quality of crops as food and the lignocellulosic composition for bioenergy applications [127] (Figure 5). As plant cell wall represents the main sink for the carbon fixed in photosynthesis, its biosynthesis is controlled by photosynthetic rate and the carbon status of the plant, being dynamically regulated as a response to environmental stresses induced by climate change [128,129].

Cell wall (CW) remodeling represents an important stress tolerance mechanism, some reports reported that significant changes in cell wall might be driven by abiotic stress to maintain growth and productivity [130,131,132]. Cellulose synthesis has been found to be affected in distinct levels by abiotic stresses such as heat, drought, and CO_2_ enrichment. An increase of 14% in cellulose content was found for Guinea grass under eTeC conditions, while 19% and 22% higher cellulose content were observed in sugarcane and *Arabidopsis*, respectively, under elevated CO_2_ concentrations [33,133,134].

In the non-cellulosic fraction of cell wall, the substitution of arabinose in the xylose backbone, indicated by the xylose: arabinose ratio was reduced for Guinea grass and coffee leaves under heat regardless of the CO_2_ concentration [135,136]. In contrast, in wheat grain arabinose substitution degree increased under heat stress [137]. The xylose: arabinose ratio is a critical parameter for biomass recalcitrance since arabinose can esterify with lignin, forming lignin–carbohydrate complexes (LCCs), which play a key role in reducing the accessibility to hydrolytic enzymes [138].

As for CW polysaccharides, the content and composition of lignin were also found to be responsive to abiotic stress induced by climate change. Under CO_2_-enriched atmosphere (eTeC) or water stress (eTwS), the lignin content of Guinea grass leaves was 16 and 17% higher for eTeC and eTwS, respectively [117,139]. Temperature is recognized as a crucial factor controlling the lignification process of plant tissues [140]. Even with the higher productivity induced by warming under well-watered conditions, lignin increase may modify industrial processes related to the use of grasses as a bioenergy source.

In contrast, an increase in levels of sinapic acid, phenylalanine, and α-Tocopherol were found under warming conditions, which was pointed to impact lignin composition by increasing the content of S-type units [34]. Freitas et al. (2020) [136] proved it, in which Guinea grass under elevated temperature conditions (warmed by 2 °C above current temperature) presented higher S lignin and S/G ratios, demonstrating a positive effect of temperature for hydrolysis since S lignin is described to easily hydrolyze upon pretreatment step. In *Eucalyptus*, drought stress disrupted lignin deposition in leaves and increased the S/G unit ratio [141].

A previous work of our group applied Carbohydrate-Binding Module (CBMs) to study the effect of climate change on the accessibility of cell wall polysaccharides to hydrolytic enzymes. We found that the higher percent glucan composition, S/G ratio, higher xylose: arabinose ratio found for Guinea grass developed under warming resulted in higher surface accessibility of cellulose and xylan, enhancing the sugar yields after biomass enzymatic hydrolysis [136].

In addition, transcriptome studies demonstrated that genes of several cell-wall-related genes like those participating in cellulose, hemicellulose, and lignin biosynthesis had transcript levels impacted by heat, drought, and elevated CO_2_ [129,142,143,144]. However, the literature data shows contrasting data on the gene/protein expression, which are highly dependent on the plant species, type, and intensity of the stresses applied. In this review, we focus on the most common proteins that are described to be up or downregulated under abiotic stresses.

Concerning cellulose, plants seem to respond to abiotic stress by upregulating the expression levels of cellulose synthases (*CesA*), sucrose synthase (*Susy*), UDP-Glc dehydrogenase (UGDh), and UDP-glucose pyrophosphorylase (UGPase) [129,131,143]. The *Susy* protein catalyzes the interconversion of sucrose into fructose and uridine diphosphate glucose (UDP-Glc) that are precursors of CW synthesis [145]. Meanwhile, *CesA*, UGDh, and UGPase are key enzymes for the biosynthesis of cellulose and other non-glucosyl CW sugars via UDP-Glc; thus, the regulation of their activities might be crucial for the synthesis of xyloglucan and other relevant matrix polysaccharides [129].

Xyloglucan is an important polysaccharide present in the hemicellulose fraction of primary cell wall of major plant species [146]. The xyloglucan endotransglucosylase/endohydrolases (XTHs) are responsible for cleaving or linking hemicellulose and cellulose inducing CW to loosen or strengthen by changing the levels of xyloglucan polymerization [147]. Along with expansins (EXP) proteins, XTH is known to have their transcriptional levels altered helping plants to adapt under abiotic stress either by inducing cell wall loosening or by reinforcing the connections between cell wall polymers [129,130,143].

Regarding lignin, a transcriptome study described in Guinea grass submitted to eTeC conditions, the downregulation of caffeic acid/5-hydroxyferulic acid O-methyltransferase 1 (COMT), an enzyme involved in lignin biosynthesis [34]. This could be a signal of a shift from lignin biosynthesis to secondary defense metabolism as a response to experienced abiotic stress and was correlated with improved maize cell wall digestibility in 30% [148]. Other enzymes such as phenylalanine ammonia-lyase (PAL) are committed to the first step of lignin synthesis through phenylpropanoid pathway, laccases, and peroxidases (e.g., apoplastic peroxidase (PRX)) responsible for monolignols polymerization, are commonly described to have their transcriptional levels as influenced by climate changes [129,143,149,150]. The adjustment in the transcription levels of lignin-related enzymes is described as an adaptive mechanism to regulate cell wall expansion and prevent water losses [143].

The alterations on cell wall polymers and in the transcription levels of cell wall-related proteins correspond to acclimation mechanisms used by plants to survive otherwise lethal abiotic stresses [129]. The results discussed in this section support the idea that tolerance mechanisms partially require cell wall-remodeling enzymes, which are involved in biosynthesis and chemical modification of cell wall polymers [131,144]. Nevertheless, most data from cell wall adjustment under climate changes come from transcriptome analysis rather than biochemical experiments. Until now, much less is known about changes in the cell wall itself, representing a challenge for a better understanding of the effects of climate change on the lignocellulosic potential for bioenergy.

Furthermore, as previously discussed in Section 7, the pretreatment step is essential for the production of fermentable sugar from lignocellulosic biomass. Nonetheless, this process alters cell wall structure, chemical composition, and association among its polymers, with the potential of influencing the climate change effect. Although warming is shown to have a positive effect on Guinea grass hydrolysis yields, the pretreatment using laccase or hydrothermal methods mitigates the climate change effects on the grass hydrolysis potential since no significant differences were found between groups grown under currently ambient conditions and eTeC treatments [133,139]. Therefore, integrating the effects of pretreatment and climate changes in lignocellulosic biomass must be investigated for potential biorefinery feedstocks.

Finally, to mitigate the fast climate changes and ensure sustainable agriculture and biomass production, innovative approaches must be developed to effectively meet the challenges and demands created by environmental changes. Advantage strategies such as metabolic engineering strategies, development of microbial consortium, macro- and micronutrient management, and the use of nutrient-coated nanoparticles have been successfully developed to improve growth and stress tolerance in plants [151].

## 10. Conclusions and Perspectives

The reduction in fossil-based energy consumption and its replacement by sustainable use of bioenergy is among the main options for fighting climate change reducing fossil carbon dioxide emissions to the atmosphere. Biomass production involves the capture of carbon from the atmosphere by the growth of biomass. However, to provide energy from biomass several processes are needed including biomass production, transport, storing, processing, conversion, and distribution. Thus, promoting the sustainable use of biomass for bioenergy requires a complete understanding of the impacts of biomass production on GHG emission, and climate change impacts on biomass production and utilization for bioenergy.

Bioenergy and food crops can be cultivated in integrated and sustainable production systems, improving land use. The stimulation of increased productivity and the use of marginal and degraded lands to dedicated energy crop production, like grasses, can reduce pressure in land use linked with bioenergy expansion and also improve carbon sequestration in soils and biomass.

Nonetheless, despite biomass availability and productivity, several factors must be considered regarding climate change. The cell wall adjustment under abiotic stress is essential in plant adaptation to environmental stresses, which directly affects the lignocellulosic potential for biorefinery. The biosynthesis of polymers and cell wall composition is dynamically regulated as a response to environmental stresses induced by climate change. Hence, since lignocellulosic biomass represents an essential alternative as energy crops, the impact of climate changes on its bioenergetic potential should be investigated. In this review, we integrate factors involved in steps for bioenergy, as pretreatment and enzymatic hydrolysis, with plant responses to climate change, showing that a complete understanding of tolerance mechanisms and climate-smart agriculture are the only way to lower the negative impact of climate changes on crop adaptation and its use for bioenergy.

## Figures and Tables

**Figure 1 biology-10-01277-f001:**
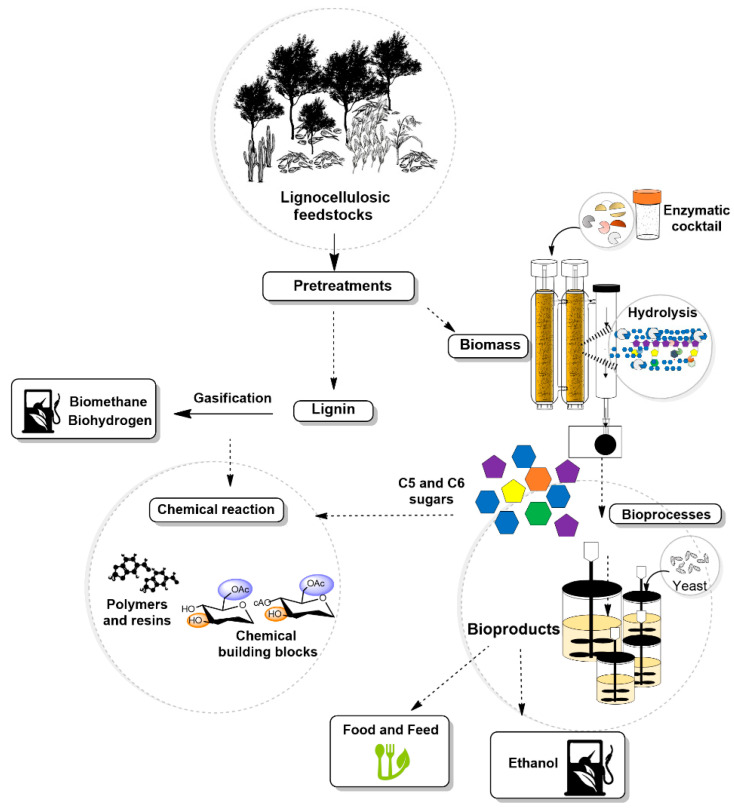
Schematic illustration of the biorefinery concept to produce fuels and chemicals.

**Figure 2 biology-10-01277-f002:**
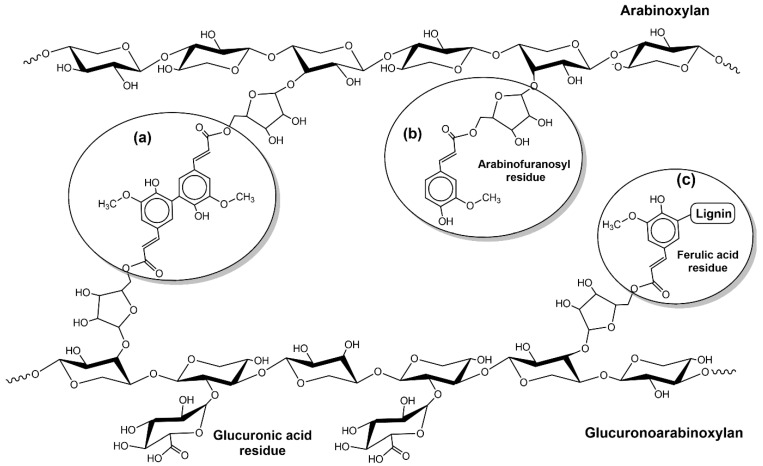
Ferulic acid esterified with arabinofuranosyl residue of glucuronoarabinoxylan (GAX) (**a**); cross-linking involving diferulic acid (**b**); ferulic acid residue attaching lignin to GAX forming lignin carbohydrate complex (LCCs) (**c**).

**Figure 3 biology-10-01277-f003:**
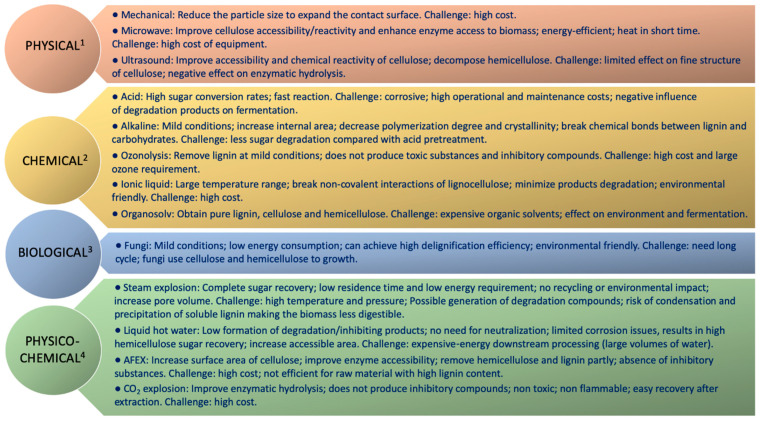
Advantages and challenges of the different pretreatment technologies.

**Figure 4 biology-10-01277-f004:**
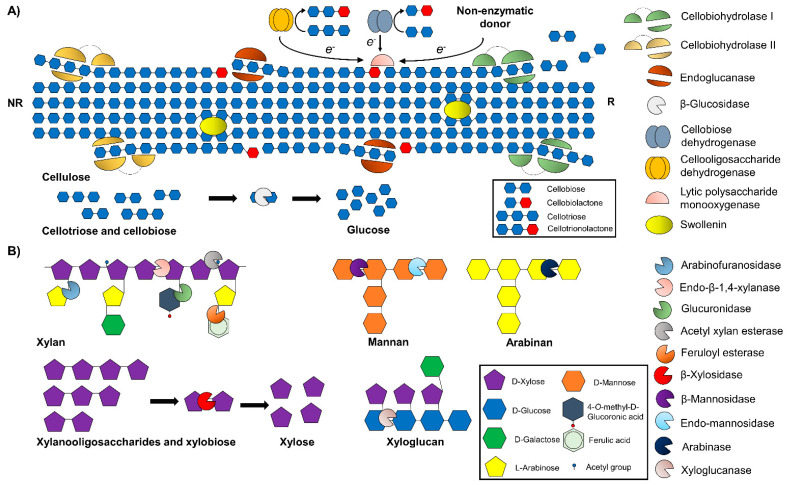
Representation of current view in cellulose and hemicellulose enzyme degradation. Enzymes from the core cellulase mixture and accessory enzymes involved in cellulose degradation. (**A**) Hemicellulolytic enzymes and accessory enzymes involved in hemicellulases debranching and degradation (**B**). NR: non-reducing end, R: reducing end.

**Figure 5 biology-10-01277-f005:**
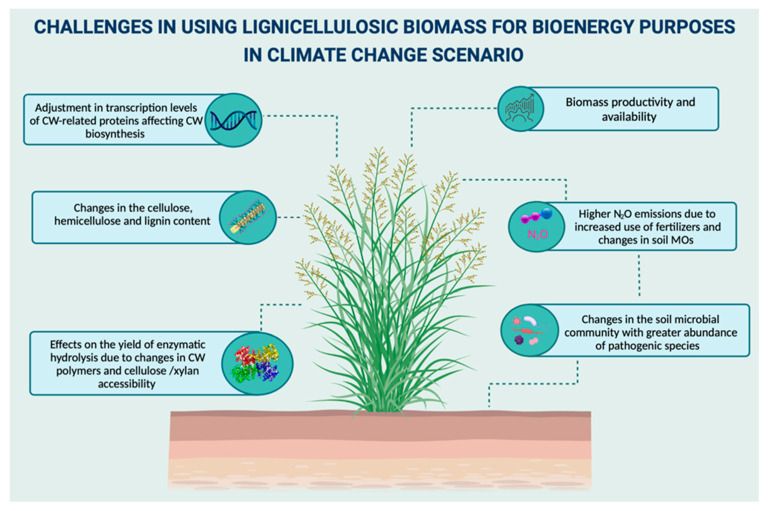
Summary of the main effects of climate change affecting the use of lignocellulosic biomass as feedstock for bioenergy production. CW: cell wall; MOs: microorganisms.

## Data Availability

No new data were collected or analyzed in this study.

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
