# Peer review of "Challenges of Biomass Utilization for Bioenergy in a Climate Change Scenario"

_biology, 2021, doi:10.3390/biology10121277_

Round 1

Reviewer 1 Report

The article entitled “Challenges of Biomass Utilization for Bioenergy in a Climate Change Scenario” by de Freitas et al. where the authors has tried to relate climate change scenario with biomass utilization for bioenergy. The article is well written but needed more changes in the organization of different sections and the approach. Authors has explained the content clearly using figures and table. 

Major changes in the format and style of the text is required before accepting in the journal

1. The format used in the article is not uniform.

2. Correct numbering of sections and subsections is missing

3. format required to b corrected in Line 69,85,90-91,112-116

4. In Line 100, 1.5C should be correctd 

5. Line 207 is not clear. Authors mention comparison with a feedstock, but no example for such feedstock was mentioned. 

6. Line 215 is not clear

7. Line 220 not clear, authors mean "commercial users" or "commercial uses"

8. Fig 1:does authors want to show that sustainable pretreatments can separate lignin from other components of biomass. All pretreatments cannot remove lignin from biomass

9. Fig 1 caption needs to be corrected

10. Atmospheric CO2 [CO2] is represented as ([CO2]) on line 120

11. Food and fedd or food and feed.

12. Sentence need correction in Lines 264 and 265.

13. Line 312 to 313 sentence need to be corrected

14. Line 340: author mention four categories of pretreatment but 3 were mentioned

15. Fig 3. Spelling errors need to be checked

16. Most of the places "mainly" is spelld as "manly"

17. Conclusion section is lenghty and give an impression of a discussion section rather than mentioning it precisely.

18. Fig 4.Instead of "glucuronidase" "glucoronidase" is written

Author Response

The text of our original manuscript that was not changed in the revised version is in standard black font. The changes are made in red font. Our replies in this document are in blue font. We have also read carefully through the text and corrected any other minor mistakes that we have found (but without highlighting them). For better understanding, we list reviewers' comments by numbering them according to their sequence in the reviewers' questions and their respective answers.

Reviewer 1:

The article entitled "Challenges of Biomass Utilization for Bioenergy in a Climate Change Scenario" by de Freitas et al., where the authors have tried to relate climate change scenario with biomass utilization for bioenergy. The article is well written but needs more changes in the organization of different sections and the approach. The authors has explained the content clearly using figures and table. 

Major changes in the format and style of the text is required before accepting in the journal

  1. The format used in the article is not uniform.

R: We thank the reviewer for the comment. We reviewed the format of the article.

  1. Correct numbering of sections and subsections is missing

R: The numbers of sections and subsections were corrected.

  1. format required to b corrected in Line 69,85,90-91,112-116

R: The format was corrected.

  1. In Line 100, 1.5C should be corrected.

R: The symbol of Celsius degree was corrected. Please see line 117.

  1. Line 207 is not clear. Authors mention comparison with a feedstock, but no example for such feedstock was mentioned. 

R: We thank the reviewer for the valuable comment. We added examples of feedstocks to make the sentence clear. Please see lines 225-226.

  1. Line 215 is not clear.

R: We tried to clarify the sentence. Please see lines 234-236

  1. Line 220 not clear, authors mean "commercial users" or "commercial uses"

R: We meant commercial uses. The text was changed. Please see line 240.

  1. Fig 1:does authors want to show that sustainable pretreatments can separate lignin from other components of biomass. All pretreatments cannot remove lignin from biomass

R: Thank you for your comment. We comply the reviewer request and changed Figure 1 for pretreatments

  1. Fig 1 caption needs to be corrected

R: Figure 1 caption was corrected.

  1. Atmospheric CO2 [CO2] is represented as ([CO2]) on line 120

R: Our purpose with ([CO2]) was to say that further in the study, the atmospheric COwould be represented as [CO2]. We tried to clarify this online 138.

  1. Food and feed or food and feed.

R: Corrected.

  1. Sentence need correction in Lines 264 and 265.

R: The sentence was corrected. Please see lines 304-305.

  1. Line 312 to 313 sentence need to be corrected.

R: The sentence was corrected. Please see lines 351-353.

  1. Line 340: author mention four categories of pretreatment but 3 were mentioned

R: Thanks for the comment. We corrected the sentence; a common was missing. Please see lines 379-380.

  1. Fig 3. Spelling errors need to be checked

R: We checked possible spelling errors in Figure 3.

  1. Most of the places "mainly" is spelled as "manly"

R: We reviewed the word manly in the text and corrected for mainly.

  1. Conclusion section is lengthy and give an impression of a discussion section rather than mentioning it precisely.

R: We thank the reviewer for the valuable comment. The conclusion section was reformulated. Please see lines 313-338.

  1. Fig 4.Instead of "glucuronidase" "glucoronidase" is written

R: Figure 4 was reviewed.

Reviewer 2 Report

Dear Authors.

I have made my comments for the authors in the pdf attached. I do not think that we need to list them again here. Please double check English.

Author Response

The text of our original manuscript that was not changed in the revised version are in normal black font. The changes are made in red font. Our replies in this document are in blue font. We have also read carefully through the text and corrected any other minor mistakes that we have found (but without highlighting them). For better understanding, we list the comments of reviewers by numbering them according to their sequence in the reviewers’ questions and their respective answers.

 Reviewer 2:

  1. hemicelluloses

R: We corrected to hemicelluloses. Please see line 70.

  1. ….other text fonts?

R: The text font was formatted.

  1. production

R: We corrected for production. Please see line 304.

  1. residues of xylose, arabinose...more exactly we discuss about xylan as poly beta 1,4 anhy-droxylopyranose...

R: We thank the reviewer for the valuable comment. The added the word residues to the text, we did not add xylan as poly beta 1,4 anhy-droxylopyranose because our intent was only to give a general idea of hemicellulose composition. Please see line 327.

  1. actually the monolignols are>para-coumaryl alcohol, coniferyl alcohol and sinapyl alcohol!

R: We comply the reviewer's request and changed the monolignols to p-coumaryl alcohol (H), coniferyl alcohol (G), and sinapyl alcohol (S). Please see line 333-334.

  1. these are actually only the aromatic part residues included in lignin struc-ture...https://link.springer.com/arti-cle/10.1007/BF00351914

R: Thanks for the comment. As the description of this section had the intent just to give essential information about lignin composition, we did not discuss the non-aromatic part of lignin. However, we added the word “mainly” to the text, to clarify that lignin is not composed only of monolignols units.

  1. three or four... figure three say otherway..

R: We corrected the categories of pretreatment, a comma as missing in the text.

  1. generally the severity factor com-bines both the effect of temperature and time (J.M. La-voie, E. Capek-Menard, H. Gauvin, E. Chornet Production of pulp from salix vinimalis energy crops using the FIRSST process Bioresour. Technol., 101 (2010), pp. 4940-4946)....in latest articles severity factor takes also into account the pH of treatment..

R: Thanks for the valuable comment. We changed the discussion by adding temperature, time, and pH as severity factors parameters. Please see lines 379-380.

  1. some lignin is also removed and dissolve during mild acidic (pH5) about 10 to 30% of the initial content is lost this way.

R: Thanks for the comment. We complement the text with this information. Please see line 395.

  1. ecological yes! but what about the costs?

R: We added to the text some discussion about the economic feasibility of enzymatic hydrolysis.

  1. what are the conditions of the study?

R: The conditions of growth for Guinea grass in the study [136] were added to the text.
